# Oligonucleotide Delivery across the Caco-2 Monolayer: The Design and Evaluation of Self-Emulsifying Drug Delivery Systems (SEDDS)

**DOI:** 10.3390/pharmaceutics13040459

**Published:** 2021-03-28

**Authors:** Jana Kubackova, Ondrej Holas, Jarmila Zbytovska, Barbora Vranikova, Guanghong Zeng, Petr Pavek, Anette Mullertz

**Affiliations:** 1Department of Pharmaceutical Technology, Faculty of Pharmacy in Hradec Kralove, Charles University, Akademika Heyrovskeho 1203, 500 05 Hradec Kralove, Czech Republic; kubackja@faf.cuni.cz (J.K.); Jarmila.Zbytovska@vscht.cz (J.Z.); vranikovab@faf.cuni.cz (B.V.); 2Faculty of Chemical Technology, University of Chemistry and Technology Prague, Technicka 5, 166 28 Prague, Czech Republic; 3DFM A/S (Danish National Metrology Institute), Kogle Alle 5, 2970 Hørsholm, Denmark; GUZE@novonordisk.com; 4Department of Pharmacology, Faculty of Pharmacy in Hradec Kralove, Charles University, Akademika Heyrovskeho 1203, 500 05 Hradec Kralove, Czech Republic; pavek@faf.cuni.cz; 5Department of Pharmacy, Faculty of Health and Medical Sciences, University of Copenhagen, 2100 Copenhagen, Denmark; anette.mullertz@sund.ku.dk; 6Bioneer: FARMA, Department of Pharmacy, Faculty of Health and Medical Sciences, University of Copenhagen, Universitetsparken 2, 2100 Copenhagen, Denmark

**Keywords:** oligonucleotide, self-emulsifying drug delivery systems, hydrophobic ion pairing, intestinal permeation enhancers, Caco-2 monolayer

## Abstract

Oligonucleotides (OND) represent a promising therapeutic approach. However, their instability and low intestinal permeability hamper oral bioavailability. Well-established for oral delivery, self-emulsifying drug delivery systems (SEDDS) can overcome the weakness of other delivery systems such as long-term instability of nanoparticles or complicated formulation processes. Therefore, the present study aims to prepare SEDDS for delivery of a nonspecific fluorescently labeled OND across the intestinal Caco-2 monolayer. The hydrophobic ion pairing of an OND and a cationic lipid served as an effective hydrophobization method using either dimethyldioctadecylammonium bromide (DDAB) or 1,2-dioleoyl-3-trimethylammonium propane (DOTAP). This strategy allowed a successful loading of OND-cationic lipid complexes into both negatively charged and neutral SEDDS. Subjecting both complex-loaded SEDDS to a nuclease, the negatively charged SEDDS protected about 16% of the complexed OND in contrast to 58% protected by its neutral counterpart. Furthermore, both SEDDS containing permeation-enhancing excipients facilitated delivery of OND across the intestinal Caco-2 cell monolayer. The negatively charged SEDDS showed a more stable permeability profile over 120 min, with a permeability of about 2 × 10^−7^ cm/s, unlike neutral SEDDS, which displayed an increasing permeability reaching up to 7 × 10^−7^ cm/s. In conclusion, these novel SEDDS-based formulations provide a promising tool for OND protection and delivery across the Caco-2 cell monolayer.

## 1. Introduction

Gene therapy offers highly specific therapeutics without significant side effects. It covers a wide variety of approaches, ranging from the replacement of malfunctioning genetic information, often in the form of a large nonviral plasmid DNA (pDNA), to the currently more investigated oligonucleotide (OND)-based therapy acting on the level of mRNA destabilization or translation [1,2]. As for OND-based therapy, synthetic 20–30-mer nucleotides connected by phosphodiester bonds form a polyanionic backbone that determines the physicochemical properties of this hydrophilic macromolecule independently of the nucleotide sequence [3]. Therefore, there should be versatility in formulating OND for various applications. Nevertheless, the nucleotide structure is unstable in the presence of nucleases. Rapid enzymatic degradation combined with insufficient cellular uptake, often followed by entrapment in endosomes, makes the delivery of OND challenging [4]. Nonviral drug delivery systems, mainly lipid-based nanoformulations, have emerged as a promising option to overcome not only these limitations but, also, to enable oral delivery [5].

Oral delivery is the preferred route of administration—in particular, in the case of local gastrointestinal diseases. Many of these pathologies are defined or accompanied by intestinal inflammation, such as inflammatory bowel disease [6,7]. Novel colloidal drug delivery systems benefit from the enhanced permeation and retention effect of the inflamed tissue, which leads to increased accumulation of nanocarriers (~100–500 nm) within the leaky inflammatory site, which is also accompanied with an altered mucus layer [8,9]. Moreover, the increased accumulation of nanocarriers enables their uptake by phagocytic immune cells invading the area affected by inflammation; their rapid elimination due to diarrhea commonly present is avoided. In addition to the size of carriers, negative surface charge and hydrophilic surface decoration are also strategies utilized for passive targeting to the inflamed intestinal tissue [9,10,11]. The local delivery of OND has been investigated using many kinds of polymeric nanoparticles [12], lipid nanoparticles [13], microencapsulated nanogel [14], thioketal nanoparticles [15] and nanoparticles-in-microspheres multicompartment systems [16].

Self-emulsifying drug delivery systems (SEDDS) are isotropic mixtures of oils, surfactants, cosurfactants and cosolvents forming oil-in-water nanoemulsions upon gentle dispersion in an aqueous environment, e.g., gastrointestinal fluids [17]. These oral lipid-based drug delivery systems were originally established to improve the bioavailability of small poorly water-soluble molecules [18]. Recently, the delivery of hydrophilic macromolecules in SEDDS has gained increasing attention in order to take advantage of its beneficial properties, such as easy upscaling, nanosize of formed droplets and protection of the loaded substance from chemical and enzymatic degradation [19]. Nevertheless, the hydrophilicity of a potential hydrophilic substance needs to be reduced, usually by hydrophobic ion pairing [19]. The hydrophobic ion pairing of nucleic acids is a method based on replacing the counterions associated with negatively charged phosphate groups with surfactants carrying the positive charge. This method leads to a lipophilicity increase of the resulting ion-paired complexes [20]. For this purpose, positively charged surfactants, such as dimethyldioctadecylammonium bromide (DDAB) and 1,2-dioleoyl-3-trimethylammonium propane (DOTAP), are frequently utilized [21,22,23,24]. In addition, these quaternary ammonium salts enhance endosomal escape so the nucleic acid can reach the cytosol [25,26].

ONDs as hydrophilic macromolecules use predominantly paracellular transport through the intestinal monolayer; therefore, their permeability under physiological conditions is very limited [27]. Oral formulations containing intestinal permeation enhancers (PEs) have been widely investigated to enhance oral delivery of hydrophilic macromolecules [28]. PEs based on medium-chain fatty acids (MCFA) have been found to enhance paracellular transport via the opening of tight junctions (TJs) in a reversible manner by interaction with the cytoskeleton [29,30]. In addition, at higher concentrations the action of MCFA might be supported by transcellular permeation as a result of cell membrane alterations [28].

Hauptstein et al. pioneered the delivery of nucleic acids in SEDDS. In their research, several hydrophobic pDNA-cationic lipid complexes were tested. Among the tested cationic lipids, the complexes with cetrimide delivered in SEDDS showed a good transfection efficiency of HEK-293 cells comparable to Lipofectin^®^, a gold standard transfection reagent [31]. Mahmood et al. improved the transfection efficiency of pDNA-cetrimide in analogical SEDDS by the incorporation of a cell-penetrating peptide and confirmed an internalization of 50-nm nanoemulsions in Caco-2 cells [32]. Both studies focused on the delivery of large pDNA into cells. In contrast to considerably larger pDNA molecules, this study focuses on the delivery of short OND sequences as emerging, highly potent therapeutics.

The aim of this study was to prepare and characterize MCFA-based SEDDS loaded with hydrophobized OND and to investigate the ability of this system to deliver OND across the intestinal Caco-2 monolayer. This formulation approach has a considerable potential to overcome OND low stability and poor permeability. In this study, firstly, hydrophobized complexes of cationic lipids (DDAB or DOTAP) and a model fluorescently labeled 20-mer OND were thoroughly described and subsequently loaded into SEDDS (Figure 1). Dispersed SEDDS (negatively charged or neutral SEDDS) were characterized in terms of size, zeta potential, lipolysis, protective effect against nucleases and OND permeability.

## 2. Materials and Methods

### 2.1. Materials

The 20-mer model 5′-6-carboxyfluorescein (FAM) labelled OND composed of random DNA nucleotides (molecular weight 6654.5 Da) was purchased from Generi Biotech (Hradec Kralove, Czech Republic). Lipoid S LPC 80^®^ (LPC) (from soybeans, containing 80.8% monoacyl phosphatidylcholine and 13.2% phosphatidylcholine) and phospholipids Lipoid S PC (from soybeans, containing 98.0%phosphatidylcholine) were provided by Lipoid GmbH (Ludwigshafen am Rhein, Germany). Labrasol^®^ (caprylocaproyl macrogol-8 glycerides), Maisine CC^®^ (glyceryl monolinoleate) and Peceol^®^ (glyceryl monooleate) were provided by Gattefossé (Saint-Priest, France). Captex 300^®^ (glyceryl tricaprylate/tricaprate) was provided by Abitec (Columbus, OH, USA) and Citrem^®^ (glycerides of citric acid) by Danisco-DuPont (Grindsted, Denmark). Dimethyldioctadecylammonium bromide (DDAB), bovine bile extract, maleic acid, porcine pancreatic lipase extract, bovine serum albumin (BSA), 2-(N-morpholino)ethansulfonic acid (MES), N-2-hydroxyethylpiperazine-N′-2-ethanesulfonic acid (HEPES) and orlistat were purchased from Sigma-Aldrich (now Merck, Darmstadt, Germany). 1,2-dioleoyl-3-trimethylammonium propane (chloride salt) (DOTAP) was purchased from Avanti Polar Lipids (Alabaster, AL, USA). Gibco^®^ Hanks’ Balanced Salt Solution (HBSS) was purchased from Thermo Fisher Scientific (Copenhagen, Denmark). S1 nuclease (Cat. No. EN0321) was purchased from Thermo Fisher Scientific (Waltham, MA, USA). Triethylamine 99.5%, glacial acetic acid 99.99%, and acetonitrile were of HPLC grade. Deionized (DI) water was purified by SG Ultraclear water systems (SG Ultra Clear™ 2002, SG Water GmbH, Barsbüttel, Germany). Hoechst 33342 and propidium iodide (PI) were purchased from Molecular Probes (Eugene, OR, USA). All other chemicals were of analytical grade and commercially available.

### 2.2. Complex Preparation

The OND and a cationic lipid (DDAB or DOTAP) were dissolved in 1 mL of Bligh–Dyer monophase consisting of chloroform:methanol:water 1:2.2:1, as previously described [33]. The monophase was separated into a two-phase system by the addition of an aliquot of 250 µL of water and chloroform. Subsequently, the mixture was vigorously vortexed for 1 min and centrifuged at 600× *g* for 5 min to achieve complete phase separation. The chloroform phase containing OND complexed with DDAB and DOTAP was collected, and the solvent was evaporated under nitrogen flow. The amount of complexed OND was quantified indirectly from the amount of noncomplexed OND in the aqueous phase. The fluorescence of FAM-labeled OND was measured in the aqueous phase at the excitation and emission wavelengths of 495 nm and 520 nm, respectively, in a microplate reader (FLUOstar Omega, BMG Labtech, Ortenberg, Germany). Complexation efficiency (CE) was calculated as follows:CE = [(initial amount of OND-amount of OND present in aq.phase)/(initial amount of OND)] × 100%(1)

Various charge ratios (+/−) of the cationic ammonium head of a cationic lipid and the anionic phosphate group of the OND backbone were tested in order to achieve efficient CE (>95%), starting at the ratio 1:1 and increasing the amount of cationic lipid. Prepared complexes were stored at −20 °C until further use [22,23].

To evaluate the changes introduced by complexation of a cationic lipid and OND, a physical mixture was prepared at the charge ratio of cationic lipid:OND 3:1. The required amount of a cationic lipid was weighed in a 4 mL screw thread glass vial (Thermo Fisher Scientific (Waltham, MA, USA). It was dissolved in a volume of chloroform sufficient to completely dissolve the lipid. Chloroform was subsequently evaporated under a nitrogen stream. The corresponding amount of OND solution was added into the glass vial, and the solvent was evaporated under the nitrogen stream. Solid lipid and OND were mixed into a physical mixture.

### 2.3. Effect of SEDDS Lipid Excipents on Complex Stability

Cationic lipid–OND complexes were prepared in the Bligh–Dyer monophase (chloroform:methanol:water 1:2.2:1) and two phases separated with an aliquot of water and chloroform, as described in Section 2.2 [23]. The effect of the lipid excipients (Captex 300, Labrasol, LPC, Citrem, Maisine CC and Peceol) in the SEDDS on the complex stability was studied. A tested SEDDS lipid excipient in the amount relevant to SEDDS composition was dissolved in chloroform and added directly to the chloroform phase containing preformed complexes. As a control, pure chloroform was added. Water:methanol solution (ratio 1:1) was added to the aqueous layer in order to keep the ratio of the top and bottom layer constant. The system was vigorously vortexed for 1 min and subsequently centrifuged at 600× *g* for 5 min. The amount of OND in the aqueous phase was determined by fluorescence, as described in Section 2.2. The CE of OND was calculated by Equation (1) and is indicated relative to the control (no added lipid).

### 2.4. Atomic Force Microscopy (AFM)

The morphology of OND and complexes was investigated by AFM imaging. AFM was performed on the Park NX20 (Park Systems, Suwon, South Korea) with tapping mode in the air, using PointProbe® Plus Non-Contact/Tapping Mode - High Resonance Frequency - Reflex Coating (known as PPP-NCHR) probes with a nominal tip radius of 7 nm. Samples deposited on freshly cleaved mica were used. For proper fixation of the nucleic acid, a divalent cation Mg^2+^ (10 mM) to electrostatically bridge the negatively charged surface of mica substrate to the anionic OND backbone was used [34,35]. Therefore, the mica was treated with 10-mM MgSO_4_ for 5 min before the OND solution was applied. Ten microliters of 1-nM OND solution was allowed to attach for 5 min, rinsed with DI water and dried under nitrogen flow. Mica surface treatment was not necessary to absorb lipophilic complexes. The samples of the complexes were initially dissolved in chloroform and subsequently diluted in methanol. Ten microliters of 0.01-nM complex solution in methanol were spin-coated on fresh mica at 1000 rpm. Results were analyzed by SPIP (Image Metrology A/S, Hørsholm, Denmark) using the Particle and Pore analysis module, which enables height and volume analyses of the particles.

### 2.5. Differential Scanning Calorimetry (DSC)

DSC was performed with the DSC 200 F3 Maia instrument (NETZSCH-Gerätebau GmbH, Selb, Germany). The samples, bulk lipids and their respective complexes were prepared as described in Section 2.2, and the physical mixtures of the same ratio between a cationic lipid:OND 3:1, prepared by mixing, were sealed in standard aluminum crucibles. The samples were cooled down to −50 °C and kept at this temperature under isothermal conditions for 10 min. The analysis was performed at a heating rate 10 K/min in the range from −50 to 150 °C. An empty sealed crucible was used as a reference. The obtained thermograms were evaluated by Proteus Software (NETZSCH-Gerätebau GmbH, Selb, Germany).

### 2.6. Attenuated Total Reflectance-Fourier-Transform Infrared (ATR-FTIR) Spectroscopy

FTIR spectra were recorded with the MB 3000-PH apparatus equipped with the MIRacle ATR sampling kit and the software Horizon MB (ABB MEASUREMENT & ANALYTICS, Zurich, Switzerland). ATR-FTIR measurement was performed on samples in a solid state placed over ZnSe ATR crystal. Typical spectra were accumulated over the spectral range 4000–400 cm^−1^ with the coaddition of 64 scans at a resolution of 16 cm^−1^ in triplicate. Reference spectra were also obtained and subtracted from the raw spectra. Spectra of blank lipids, complexes at different charge ratios, physical mixtures of cationic lipid:OND 3:1 and OND were obtained. In order to observe complex-specific bands, spectra of the respective cationic lipids and OND were subtracted from the spectra of a complex.

### 2.7. Preparation of SEDDS Formulations

The composition of the utilized SEDDS is depicted in Table 1, as described by Ramakrishnan Venkatasubramanian et al. (data not published, manuscript in preparation).

The structure of utilized SEDDS excipients is shown in Figure 2. All excipients were weighed in a glass vial and homogenized by stirring with a magnetic bar at 37 °C overnight. Two SEDDS were used, each differing in the surface charge: negatively charged Citrem SEDDS and neutral Standard SEDDS. The chosen SEDDS were reported to differ only in the surface charge while maintaining a comparable size suitable for targeting in leaky inflammatory lesions.

The SEDDS were loaded with the OND complexed with a cationic lipid, the cationic lipid itself or orlistat. The loaded substance was weighted into a vial. Subsequently, the amount of SEDDS was added to reach the required concentration of 100 nmol of a complex per 1 g of SEDDS, 3.8 mg of DDAB or 4.2 mg of DOTAP per 1 g of SEDDS. Orlistat inhibits the activity of gastrointestinal lipases, and in this study, it was used at a concentration of 0.25% (*w*/*w*), prepared as 2.5 mg of orlistat dissolved in 1 g of SEDDS. Loading was achieved by dissolution of the loaded substance in the SEDDS under stirring with a magnetic bar at 37 °C overnight. The loaded substances and their respective concentrations utilized throughout the study are summarized in Table 2. The amount of loaded cationic lipids was equivalent to the amount present in the ion-paired complexes.

### 2.8. Characterization of Dispersed SEDDS

The dispersions of both Citrem and Standard SEDDS were characterized in terms of size and polydispersity index (PdI) by dynamic light scattering; zeta potential was measured by laser Doppler electrophoresis using Zetasizer Nano ZS (Malvern Instruments, Worcestershire, UK). The dispersions of blank and complex-loaded SEDDS were dispersed 1:100 (*w*/*w*) in DI water and 10-mM MES-HBSS buffer (pH 6.5) immediately before the experiment and evaluated at 37 °C. Data were collected in triplicate.

### 2.9. Dynamic In Vitro Lipolysis of SEDDS

Dynamic one-compartment in vitro lipolysis was carried out under human fasted-state conditions, as previously described [38,39,40]. Nonloaded Citrem and Standard SEDDS both without and with the addition of orlistat 0.25% (*w*/*w*) were tested. The addition of orlistat, a lipase inhibitor, into the SEDDS composition has previously been described to inhibit the digestion of SEDDS [41]. SEDDS (0.5 g) was added into a temperature-controlled vessel containing fasted-state simulated intestinal fluid (FaSSIF) used as the lipolysis medium (2.95-mM bovine bile, 0.26-mM soy phospholipids, 2.0-mM maleic acid, 2.0-mM Tris, 50.0-mM NaCl and 1.4-mM CaCl_2_, adjusted to pH 6.5). The digestion of the lipids was initiated by the addition of 5 mL of pancreatin solution with a final activity of 500 USP units/mL. The pancreatin solution was prepared by combining the appropriate amount of porcine pancreatin and adding intestinal lipolysis medium. The resulting suspension was vortexed and centrifuged (7 min, 6500× *g*). pH 6.5 was maintained by an automated pH stat (Metrohm Titrino 744, Tiamo Version 1.3, Herisau, Switzerland), as the released free fatty acids were continuously titrated by 0.4-M NaOH. The lipolysis was performed for 60 min in three independent experiments (*n* = 3).

### 2.10. Cryogenic Transmission Electron Microscopy

The structures formed upon the dispersion of the SEDDS 1:100 (*w*/*w*) in FaSSIF (Biorelevant, London, UK) at 37 °C were observed by cryogenic transmission electron microscopy (cryo-TEM). Individual samples (3.5 µL) were loaded on freshly glow-discharged TEM grids (Quantifoil, Cu, 200 mesh, R2/1) inside the Thermo Scientific Vitrobot Mark IV (Thermo Fisher Scientific, Waltham, MA, USA) and plunged frozen into liquid ethane. The Vitrobot chamber was kept at 4 °C and 100% relative humidity during the entire process. Each sample was incubated for 30 s on the grid and blotted for 5 s prior to vitrification. The grids were subsequently mounted into the Autogrid cartridge and loaded into the Thermo Scientific Talos Arctica transmission electron microscope (Thermo Fisher Scientific, Waltham, MA, USA). The microscope was operated at 200 kV, and cryo-TEM data was collected on the Thermo Scientific Falcon 3EC direct electron detector operating in charge integration mode. The micrographs were acquired using SerialEM software at magnifications corresponding to the calibrated pixel sizes of 22.16 Å/px and 3.15 Å/px, respectively. Each micrograph collected at 3.15 Å/px comprised an overall dose of 10 e/Å^2^.

### 2.11. Degradation by Nucleases

In order to assess the ability of the SEDDS to protect the encapsulated OND complexes against degradation by nucleases, formulations were incubated with S1 nuclease (100 U/µL), an enzyme that specifically degrades single-stranded nucleic acids. Stability studies were carried out in the following formulations: aqueous solution of naked OND, aqueous dispersion of complexed DDAB-OND in the SEDDS and DOTAP-OND in the SEDDS and dispersion of the nonloaded SEDDS in an aqueous solution of naked OND. The SEDDS were dispersed immediately before the experiment.

The S1 nuclease assay (Thermo Fisher Scientific, Waltham, MA, USA) was performed according to the manufacturer’s instructions. Briefly, equivalents containing 1.2 µg of OND were incubated with 12 U of S1 nuclease in the presence of the reaction buffer pH 4.5 at 25 °C. As controls, the same formulations were incubated in an acetate buffer of pH 4.5. The reaction was terminated after 30 min by the addition of 0.5-M EDTA and incubation for 10 min at 70 °C. Nondegraded intact OND was precipitated via ethanol precipitation. From each formulation, 60 µL was taken, and 1.3 µL of 5%NH_4_OH, 30 µL of 7.5-M ammonium acetate and 400 µL of 96% ethanol were added into the samples, which were kept at −80 °C for at least one hour. Following this, the samples were centrifuged at 15,000× *g* for 30 min at 4 °C. Intact OND was pelleted and dissolved in 500 µL of 100-mM triethylammonium acetate (TEAA). The samples were subjected to HPLC analysis. Additionally, 100 µL of supernatant was diluted with 500 µL of 100-mM TEAA.

Thirty microliters of the samples were analyzed using the Waters Symmetry C18 (3.5 µm; 4.6 × 75 mm) HPLC column (Waters Corporation, Milford, MA, USA). The mobile phase consisted of 2 eluents, eluent A of 100-mM TEAA (pH 7.0) and eluent B of acetonitrile, with a linear gradient starting from 1 min at 10% to 30% B in 4 min. Thirty percent B eluent was maintained for 1 min, then decreased to 10% B in 1 min and maintained at this level for 4 min before analysis of the following sample. The flow rate was 1 mL/min. FAM-labeled OND was detected by fluorescent detection excitation/emission 495 nm/520 nm according to the procedure described previously [42]. Each formulation was tested in triplicate. The percentage of intact OND was calculated as a ratio of area under the curve (AUC) of protected OND found in the pellets after incubation with S1 nuclease and pelleted OND recovered from the respective formulations, as described by Equation (2).
intact OND = [AUC of pelleted OND (formualtation + S1 nuclease)/AUC of pelleted OND (formulation)] × 100%(2)

### 2.12. Caco-2 Cell Monolayer Transport Study

Caco-2 cells, a model of human enterocyte epithelial cells, were obtained from the American Type Culture Collection (ATCC) (Manassas, VA, USA). The cells were seeded on T12 filter inserts (pore size 0.4 mm, 1.13 cm^2^ growth area, Corning, Sigma–Aldrich, Copenhagen, Denmark) at a final density of 1 × 10^5^ cells/cm^2^. The cells were cultured for 20–23 days before the transport study was conducted.

Transport experiments were conducted on a horizontal shaker at 37 °C using a pH gradient mimicking physiological conditions. A volume of 500 µL of 10-mM MES-HBSS buffer (pH 6.5, 0.05% *w/v* BSA) was used on the apical side and 1000 µL of 10-mM HEPES-HBSS buffer (pH 7.4, 0.05% *w/v* BSA) on the basolateral one for 120 min. The experiment was initiated by applying 500 µL of a formulation on the apical side. The following formulations were tested: OND solution, DDAB-OND in the SEDDS, DOTAP-OND in the SEDDS, DDAB in the SEDDS dispersed in OND solution, DOTAP in the SEDDS in dispersed OND solution and nonloaded blank SEDDS dispersed in OND solution for both the Citrem and Standard SEDDS. The SEDDS were dispersed 1:100 (*w*/*w*) shortly before the experiment in a MES-HBSS-based buffer preheated to 37 °C. All formulations finally contained 1 nmol/mL of OND.

Transepithelial electrical resistance (TEER) was monitored with the EVOM Epithelial Voltohmmeter (World Precision Instruments, East Lyme, CT, USA). Before and after the experiment, the cell culture cup Endohm chamber was used at 25 °C in MES-HBSS and HEPES-HBSS applied apically and basolaterally, respectively. In addition, TEER values were also monitored using chopstick electrodes throughout the experiment at 30, 60, 90 and 120 min at 37 °C.

Samples of 100 µL were taken from the basolateral side at predetermined time points (15, 30, 45, 60, 90 and 120 min) and replaced with the same volume of fresh prewarmed HEPES-HBSS buffer. The withdrawn samples were analyzed for fluorescently labeled OND by measuring fluorescence excitation/emission wavelengths at 495/520 nm in a microplate reader (FLUOstar Omega, BMG Labtech, Ortenberg, Germany). The apparent permeability coefficient (P_app_), and transported OND accumulated basolaterally across the Caco-2 monolayer were calculated using the following Equations (3) and (4).
P_app_ = J/C_0_ = Q/(C_0_ × A)(3)
transported OND = [Q/(0.5 × C_0_)] × 100%(4)
where J indicates the flux over a steady-state time period (pmol/s/cm), and C_0_ stands for the initial concentration of OND in the apical compartment (pmol/mL). Q is the accumulated mass of OND transported across the monolayer on the basolateral side (pmol) of the area A (cm^2^) of membrane filters.

### 2.13. In Vitro Cytotoxicity Study

Cytotoxicity of the formulations was investigated as the loss of cell membrane integrity accompanied by release of a cytoplasm enzyme lactate dehydrogenase (LDH). The cytotoxicity study was conducted by the Cytotoxicity Detection Kit (LDH) (Roche, France) according to manufacturer’s instructions after 120 min of the transport experiment in the Caco-2 cells. Briefly, a 40-µL sample of apical medium was diluted 5-fold by DI water. Fifty microliters of the diluted sample were pipetted into a clear 96-well multiwell plate in duplicates, and 50 µL of LDH reaction mixture was added. The 96-well plate was incubated for 30 min at 37 °C in a horizontal shaker in the dark. Subsequently, the absorbance was measured at 492 nm.

Results are depicted as cell viability (%). Relative cytotoxicity (%) was calculated according to the manufacturer’s instructions, and the resulting values were deducted from 100% to obtain cell viability (%).

### 2.14. Uptake Study

Caco-2 cells were seeded onto confocal dishes (Nunc^TM^ 4-well Rectangular Dishes, Thermo Fisher Scientific, Waltham, MA, USA) at a density of 10^5^ cells/cm^2^ and were cultured for 7 days until 100% confluence was reached. The experiment was initiated by the addition of 0.5 mL of a test sample. In order to mimic the intestinal environment, the test samples were prepared in 10-mM MES-HBSS buffer (pH 6.5, 0.05% *w/v* BSA). The uptake of OND was evaluated by testing the following samples: OND solution, nonloaded SEDDS dispersed in OND solution, DDAB-OND in SEDDS and DOTAP-OND in SEDDS dispersed in MES-HBSS buffer. The experiment was carried out for both Citrem and Standard SEDDS, and SEDDS were dispersed at 1:1000 (*w*/*w*). The cells were incubated with the samples for 2 h at 37 °C and 5% CO_2_ without shaking. Fifteen minutes before the termination of the incubation, cell nuclei were stained by the addition of 10 µL of Hoechst 33342 (1 mg/mL). At the same timepoint, 10 µL of propidium iodide (PI) (0.1 mg/mL) was added to detect dead cells.

Immediately after incubation, cells were washed twice with prewarmed Opti-MEM® medium (Thermo Fisher Scientific, Waltham, MA, USA). Images were acquired with the Nikon Ti-E (Nikon, Minato, Japan) epifluorescence microscope equipped with a cooled scientific CMOS camera (AndorZyla 5.5; Andor Technology, Belfast, UK) and LED fluorescence source (CoolLED pE-300; CoolLED, Andover, UK). DAPI, FITC and Cy3 filter sets were used for image Hoechst, FAM-labeled OND and PI, respectively. Three representative images of each sample were taken using NIS Elements AR 4.20 software (Laboratory Imaging, Prague, Czech Republic).

### 2.15. Statistical Analysis

Statistical analysis was performed using Graph Prism 8 (GraphPad Software, San Diego, CA, USA). One-way ANOVA followed by Tukey’s post-hoc test was carried out for the evaluation of size and zeta potential of the dispersed SEDDS. Two-way ANOVA followed by Tukey’s test was performed to evaluate lipolysis data and in vitro data. One-way ANOVA followed by Dunnett’s post-hoc test was used, as the impact of the other lipids on the complex stability was evaluated as well, as in the case of the in vitro data. Data are presented as mean ± standard deviation (SD). Significant difference was considered at *p* < 0.05.

## 3. Results

### 3.1. Complex Preparation

First, complexes of a cationic lipid and 20-mer fluorescently labeled OND were prepared using the Bligh–Dyer method [23,33]. Various charge ratios between a quaternary ammonium head group of a cationic lipid (N^+^) and a nucleotide phosphate group in the backbone of 20-mer OND (PO_2_^−^) were investigated. Upon complexation, no visible precipitated material in the aqueous phase nor at the interface was observed. The CE of DDAB-OND and DOTAP-OND are depicted in Table 3. Efficient CE (>95%) was achieved for the charge ratio of cationic lipid:OND 3:1, which equals the molar ratio 60:1. Moreover, at this charge ratio, there was no significant difference in CE between DDAB and DOTAP. It is noteworthy that DOTAP also enables a more efficient hydrophobization of OND at a lower charge ratio than DDAB.

### 3.2. Effect of SEDDS Lipid Excipents on Complex Stability

Next, the impact of the SEDDS excipients on the destabilization of the complexes was tested. The tested excipients included anionic Citrem, zwitterionic LPC and neutral Captex 300, Labrasol, Maisine CC and Peceol. The results from the combination of LPC and DOTAP-OND could not be determined as emulsification occurred, disabling phase separation under the method’s conditions.

Only anionic Citrem had a negative impact on the stability of both DDAB-OND and DOTAP-OND (*p* < 0.001) (Table 4). Zwitterionic LPC and neutral excipients had no impact on the complex stability. These results are in accordance with the observations by Wong et al. [23].

### 3.3. Atomic Force Microscopy (AFM)

OND and the cationic lipid-OND complexes were imaged by AFM in order to compare their size and morphology. It is important to note that AFM imaging only provides information on surface immobilized structures, and the absolute accuracy is limited by the physical convolution between the tip (radius around 7 nm) and the sample. Nevertheless, AFM topography images are useful for comparing the sizes and for providing a direct observation of the morphology of the OND and complexes on the surfaces.

Figure 3A shows a topographic image of OND molecules dispersed on mica. The molecules were compacted instead of stretched out over the mica substrate, suggesting a potential effect of Mg^2+^ (a bridging agent used for deposition on the mica substrate) on the molecular conformation [34]. An analysis of height cross-sections indicates a step height of around 1 nm, which is in agreement with the size of a single strand DNA OND, as a double helix of DNA has a diameter of 20 Å (=2 nm) [43,44].

The AFM topographic image of DDAB-OND complexes (Figure 3B) reveals a mixture of nanoparticulate complexes and membrane patches which are presumably the result of collapsed lipid capsules during sample drying. The height of the membrane patches is 4 to 5 nm. In contrast, the topographic image of the DOTAP-OND complexes (Figure 3C) shows mostly homogenous and intact particles. Average height of the particles was 15.1 ± 6.3 nm (*n* = 167) and 20.9 ± 3.5 nm (*n* = 70) for DDAB-OND and DOTAP-OND, respectively. The actual height of both complexes in solution should be higher, as significant deformation is expected during drying and immobilization on the substrate during sample preparation. The spherical lipid capsule collapsed into a shape of a flattened disc, having maintained its volume. Using the reported particle volume from SPIP software, the diameter of the particles was calculated for the original thermodynamically favoured spherical shape. The expected diameters of the spherical complexes are in the range of 31.6–63.4 nm for the DDAB-OND complexes and 75.4–128.2 nm for the DOTAP-OND complexes.

### 3.4. Differential Scanning Calorimetry (DSC)

Figure 4 depicts thermograms of the bulk lipids (DDAB/DOTAP), physical mixtures (mix DDAB/DOTAP+OND) and complexes (DDAB/DOTAP-OND), both at the charge ratio 3:1. The bulk lipids showed melting peaks at 88.5 °C and 6.3 °C, respectively, for DDAB and DOTAP. This difference is a function of the chemical structure of the cationic lipids. Double bonds in DOTAP’s aliphatic C18 chains lead to more loose packing, which provide more fluidity, resulting in a lower melting point [45].

For DDAB, both the physical mixture and the complex revealed several melting peaks, and they melted at lower temperatures (Figure 4A). The melting peaks were found at 56.5, 71.6 and 86.7 °C for the physical mixture and at 39.0, 68.3 and 85.0 °C for DDAB-OND. Evidently, the complex shows different thermotropic behavior compared to the physical mixture.

Upon mixing and complexing DOTAP with OND, the melting temperature decreased to 1.3 and 3.5 °C, respectively (Figure 4B).

### 3.5. ATR-FTIR Spectroscopy

In the region of the FTIR spectra depicting changes in base pairing (1800–1500 cm^−1^), OND bands of 1643 cm^−1^and 1605 cm^−1^ were observed (Figure 5A,B). Upon complex formation, these bands moved to the higher wavenumbers of 1659 cm^−1^ and 1697 cm^−1^. The positive shift in this FTIR region suggests the disintegration of the hydrogen bonding network involved in base pairing [24,46]. Although the model OND is single-stranded, partial base pairing could arise.

The list of complex-specific bands is summarized in Table 5. The OND peak at 1211 cm^−1^ shifted positively to 1257 cm^−1^ in both complexes but remained at the same position (1211 cm^−1^) in the mixture. A new band appeared at 1095 cm^−1^ and 1088–1080 cm^−1^ in DDAB-OND and DOTAP-OND, respectively. These bands characterize P=O vibrations. The peak at 1018 cm^−1^, typical for the P-O-C bond, gained more intensity upon complexation. This suggests an increase in the vibrational frequencies of anionic PO_2_ in the OND backbone as a result of an interaction with the cationic head of a lipid.

An intense peak at 802 cm^−1^ was observed in all complexes, suggesting a change in the conformation of deoxyribose (Figure 5A,B). Together with this intense peak, a broader peak at 864 cm^−1^ in the case of DOTAP-OND suggests an N-type (C3′-endo) sugar puckering mode [46]. For DDAB-OND, no other markers of the N-type puckering mode existed. This could be due to the DDAB peak occurring at a similar wavelength (887 cm^−1^), resulting in an interference with other N-type marker bands. In the physical mixtures, there were less intensive and broader peaks at 879–887 cm^−1^ and 840 cm^−1^ in DDAB-OND and DOTAP-OND, respectively. These bands are characteristic of S-type sugars [46]. In contrast, the bands found in OND showed a rather broad shoulder peak at 818 cm^−1^. The position of this broad shoulder peak is unspecific and might suggest the presence of both sugar puckering modes.

The spectra of the cationic lipids carried several specific bands (Figure 5C,D). The ester group of DOTAP gave rise to a peak at 1744 cm^−1^ [47,48] that also remains in the same position after complexation with OND. A weak band of around 1489 cm^−1^ characteristic of the trimethylammonium headgroups [47] was not observed in any of the cationic lipids. This weak band is likely overlapped by the CH_2_ scissoring deformation at 1466 cm^−1^ seen in both lipids. In the formed complexes, this peak did not change its position, which correlates with the behavior of lipid peaks at higher wavelengths in the C-H stretching region. In DDAB, the symmetric (2849 cm^−1^) and asymmetric (2916 cm^−1^) CH_2_ vibrations and, similarly, the symmetric (2854 cm^−1^) and asymmetric (2924 cm^−1^) CH_2_ vibrations of DOTAP remained at the same vibrational frequency upon complexation at all tested ratios [47], which indicated no changes in the lipid chain arrangement.

### 3.6. Characterization of Dispersed SEDDS

Both Citrem and Standard SEDDS preconcentrates were clear oily formulations that immediately dispersed into milky emulsions both in DI water and MES-HBSS buffer (pH 6.5). The dispersion in DI water was evaluated in order to assess properties of the formulations and in MES-HBSS to mimic the pH, as well as a more complex environment of in vitro testing media, as shown in Table 6.

The size (z-average) was found between 180 and 270 nm for both SEDDS in both investigated media, with the PdI ranging from 0.125 to 0.360. In the case of the Citrem SEDDS, there was no significant difference in size across the formulations and media. The Standard SEDDS loaded with DOTAP-OND formed significantly larger nanostructures in the DI water than nonloaded Standard SEDDS (* *p* < 0.05). In MES-HBSS buffer, a similar phenomenon was observed; in this case, the DDAB-OND-loaded Standard SEDDS created larger structures than the nonloaded formulation (*** *p* < 0.001).

The zeta potential was significantly influenced by the composition of SEDDS and the dispersing media. In DI water, both nonloaded SEDDS showed a negative surface charge: −35.5 ± 0.6 and −5.2 ± 2.1 mV for Citrem and Standard SEDDS, respectively. After loading, the surface charge of the Citrem SEDDS increased to about −25 mV (^###^
*p* < 0.001) and turned positive in the loaded Standard SEDDS to about 13 mV (^###^
*p* < 0.001). Both complexes increased the zeta potential of the nonloaded Citrem and Standard SEDDS to the same extent. The Citrem and Standard SEDDS in MES-HBSS buffer (pH 6.5) showed no significant differences within the same kind of SEDDS among the nonloaded and loaded SEDDS; about −9.5 and 0 mV for Citrem and Standard SEDDS, respectively.

### 3.7. Dynamic In Vitro Lipolysis of SEDDS

Human intestinal digestion of SEDDS under fasted-state conditions was simulated by dynamic in vitro lipolysis. Lipolysis of the SEDDS was performed with the nonloaded Citrem and Standard SEDDS, as well as both SEDDS with orlistat, in order to evaluate the degree of digestion (Figure 6). The amount of released free fatty acids was the same for both SEDDS. The addition of orlistat significantly decreased lipolysis to ~10% of the formulations with no orlistat (*** *p* < 0.001). No significant difference was observed between the Citrem and Standard SEDDS upon the addition of orlistat. Moreover, lipolysis of DDAB in SEDDS and DOTAP in SEDDS was performed. None of cationic lipids had any impact on the digestion process (data not shown).

### 3.8. Cryo-TEM

SEDDS were dispersed in FaSSIF at 37 °C in the ratio 1:100 (*v*/*v*). The final concentration of OND in dispersion of loaded SEDDS was 1 nmol/mL. In total, 69 images were evaluated, with one representative image per formulation presented in Figure 7. Figure 7A–C depicts the dispersed Citrem SEDDS, nonloaded, DDAB-OND and DOTAP-OND-loaded, respectively. Figure 7D–F shows the Standard SEDDS, nonloaded, DDAB-OND and DOTAP-OND-loaded, respectively.

All formulations dispersed into nanosized structures. Predominantly, spherical uni- and multilamellar vesicles (darker and more pronounced borders) were observed, while oil droplets occurred less frequently (darker spheres with brighter borders). Wu et al. described similar differences between vesicles and oil droplets in a soybean oil-in-water emulsion stabilized by egg yolk lecithin [50]. The nanostructures varied in size from 100 to 300 nm. In addition, cryo-TEM detected more complex spherical and elongated vesicular structures (such as in Figure 5A,B,D), usually of larger sizes (>300 nm). No apparent differences between the six different SEDDS could be observed.

### 3.9. Protection Against Nuclease Degradation

The S1 nuclease completely degraded OND in an aqueous solution of naked OND (data not shown). Dispersions of both nonloaded SEDDS in the aqueous solution of naked OND showed the same pattern. This confirmed that neither the composition of Citrem SEDDS nor Standard SEDDS interfered with the activity of the enzyme.

If OND becomes cleaved by a nuclease, short OND fragments are detected in the supernatant that do not precipitate into pellets. The amount of OND that remain intact upon contact with the S1 nuclease was detected in the pellets and is shown in Table 7. There was a significant difference in the amount of OND protected by the Citrem and Standard SEDDS, as the Standard SEDDS showed about 3.5-fold higher protection (*p* < 0.001). On the other hand, no difference was observed between DDAB-OND and DOTAP-OND in any of the SEDDS.

### 3.10. Caco-2 Cell Monolayer Transport Study

Several parameters of the prepared formulations were investigated in vitro in the Caco-2 monolayer. As an indicator of permeation enhancement, the TEER values were determined by an Endohm chamber at 25 °C before and after the experiment. After 120 min, the TEER decreased significantly if incubated with the Citrem or Standard SEDDS. No changes occurred after incubation with OND solution or the MES-HBSS buffer serving as the control. There were no significant changes among the formulations based on the Citrem SEDDS nor Standard SEDDS, indicating that the TEER changes can be attributed to the two SEDDS. Nevertheless, the extent to which the two SEDDS decreased the TEER differed significantly. The final TEER at 120 min of the Citrem SEDDS and Standard SEDDS-treated wells represented 66% ± 10% and 28% ± 4% of the initial TEER values, respectively (*p* < 0.01) (Figure 8A). In addition, the TEER values seem to be influenced differently by the Citrem and the Standard SEDDS.

This aspect was further evaluated by chopstick electrodes throughout the experiment at 37 °C starting from 30 min of the experiment and related to this time point. The cells incubated with the Citrem SEDDS had already reached the final TEER values by this time point, with the values remaining unchanged until the end of the experiment. This contrasts with the Standard SEDDS, in which TEER after 60 min reached a plateau at 80% of the TEER at 30 min (Figure 8C,D).

Figure 9A,B shows the cumulative OND transport into the basolateral compartment. At 120 min, both SEDDS displayed a significant increase of transported OND compared to the OND solution (Citrem SEDDS ^#^*p* < 0.05 and Standard SEDDS ^##^
*p* < 0.01 and ^###^
*p* < 0.001). In the case of Citrem SEDDS (Figure 9A), no difference was observed between the two complexes. For the Standard SEDDS (Figure 9B), DDAB and DOTAP in the SEDDS and, also, the nonloaded SEDDS enabled the transport of higher amounts of OND than DDAB-OND and DOTAP-OND in SEDDS over 120 min (* *p* < 0.05). The total transported mass of OND after 120 min is summarized in Table 8.

A linear steady state of the flux curves was taken as the basis for the calculation of P_app_. In order to investigate in detail changes in permeability across the Caco-2 monolayer during 120 min, the time frame of the experiment was divided into three intervals, i.e., Δ15–45 min, Δ30–60 min and Δ60–120 min, with the P_app_ values calculated for each interval. As can be seen in Figure 9C, the P_app_ was stable for all Citrem SEDDS formulations at all time intervals. OND solutions containing dispersions of the Citrem SEDDS with or without DDAB or DOTAP already showed a significant increase at the interval of Δ30–60 min relative to the OND solution. At the final interval of Δ60–120 min, all Citrem SEDDS significantly improved the permeability of OND relative to the respective time intervals of the OND solution (*p* < 0.05). In contrast, for the Standard SEDDS, the P_app_ increased over time for all formulations (*p* < 0.05) (Figure 9D). Only the OND solution with dispersions of DDAB and DOTAP in the Standard SEDDS already increased the P_app_ at the initial time interval Δ15–45 min. At the following interval of Δ30–60 min, the P_app_ of DOTAP-OND in the Standard SEDDS and the nonloaded Standard SEDDS was also enhanced significantly. Finally, at the interval of Δ60–120 min, all formulations of the Standard SEDDS showed an increase in P_app_ relative to the OND solution.

### 3.11. In Vitro Toxicity Study

The viability of the Caco-2 cells was evaluated by an LDH assay after 120 min of treatment with all SEDDS (Figure 8B). The OND solution was confirmed to be nontoxic. For both the Citrem and Standard SEDDS, there were no significant differences between the applied SEDDS formulations, which suggests that the main contribution to the cytotoxicity is from the surfactants in SEDDS and not the cationic lipids or the complexes. All Citrem SEDDS had viability values close to 100%, while the Standard SEDDS had a decreased viability to about 70% (* *p* < 0.05 and ** *p* < 0.01). These data might indicate increased cell stress due to compromised integrity of the cell membrane.

### 3.12. Uptake Study

The uptake of the fluorescently labeled OND formulated into SEDDS was examined in confluent Caco-2 cells using fluorescent microscopy. Figure 10 shows merged images: blue Hoechst staining cell nuclei, red PI staining dead cells and green FAM-labeled OND. Images depicting staining with a single dye can be found in the Appendix A.

Similar to the OND solution (Figure 10B), no OND was observed in living Caco-2 cells upon incubation with the Citrem or Standard SEDDS or when OND was administrated in the aqueous phase (nonloaded SEDDS; Figure 10C,D) or complexed inside the SEDDS (DDAB-OND and DOTAP-OND; Figure 10E,F and Figure 10G,H, respectively). The green signal of FAM-labeled OND was observed only in combination with the red signal indicating that OND was localized only inside dead Caco-2 cells.

## 4. Discussion

### 4.1. Preparation of Cationic Lipid-OND Complexes

Cationic lipid-DNA complexes have been shown to be a less immunogenic but, also, a less efficient alternative to viral vectors for the delivery of nucleic acids [4]. These macromolecules are often formulated as lipoplexes with preformed liposomes to enhance the cell transfection of large DNA molecules (several kb) [51], as well as shorter siRNA (tenths of bp) [52], for parenteral delivery to the systemic circulation. In this study, we prepared hydrophobic complexes consisting of cationic lipids and OND in order to load them into SEDDS, a well-established lipid-based oral delivery system. The presented experiments were conducted with a model nonspecific fluorescently labeled OND. Nevertheless, the structural features involved in the key procedures, such as the preparation of ion pairs with cationic lipids and, thus, hydrophilicity reduction, are applicable for potential therapeutic OND sequences. Therefore, the easily detectable fluorescently labeled model OND was used to investigate this innovative formulation process.

Monovalent cationic lipids with two lipophilic chains, DDAB and DOTAP, are known to be less toxic than their single-tailed counterparts [53]. Both aliphatic chains of DDAB are saturated, unlike the chains of DOTAP, which have a double bond in the C9 position. This influences their physicochemical properties; DDAB exists in the gel state and DOTAP in the liquid crystalline state at room temperature [54]. In the present study, it was found that the more flexible DOTAP chains seem to be more easily assembled around the OND core, leading to CE > 95% already at the charge ratio 2:1 (N^+^:PO_2_^−^). At the ratio 3:1 (N^+^:PO_2_^−^), there was no significant difference between the CE of the cationic lipids (Table 3). At all employed ratios, no lipid was observed at the interface during the Bligh–Dyer extraction, suggesting that most of the lipid contributed to the complexation. The ability of DDAB to create a more ordered structure was shown also by DSC, as several melting peaks of crystalline assembles were observed (Figure 4). The more ordered structure and, consequently, tighter molecular packing could explain the smaller size of the DDAB-OND complexes, as detected by AFM (Figure 3).

The formation of complexes between a cationic lipid added as a monomer or in micellar form and plasmid DNA (pDNA) has already been described and well-characterized [23,55]. In both published preparation methods, a sufficient recovery of pDNA was reported at the charge ratio 1:1. The recovery process was more efficient (>90%) if a lipid was added in excess of 1.5:1 (N^+^:PO_2_^−^) [55]. For shorter siRNA, the need of higher lipid/OND ratio was reported [52,56]. It is widely accepted that the mechanism of complex formation is common for any type of nucleic acid and cationic lipid [57]. In other words, both components of the complex self-assemble into a core–shell structure based on the ion pairing—more specifically, primarily on electrostatic interactions between the cationic headgroup of a lipid and anionic phosphate of the nucleic acid backbone.

The ATR-FTIR analysis (Figure 5 and Table 5) showed complex-specific bands that occurred mostly in the region characteristic of changes in the phosphate-deoxyribose backbone and confirmed the interactions with the OND backbone in the complex formation process. These interactions could not be observed in physical mixtures prepared with the same cationic lipid:OND ratio. Previous studies showed that electrostatic interactions help to form inverted micelles, encapsulating the nucleic acid in the core surrounded by the shell of the cationic lipid [23]. This is in agreement with the AFM topography images showing collapsed lipid shells around the cores, as observed in both the DDAB and DOTAP complexes (Figure 3).

### 4.2. Loading the Complexes into Citrem and Standard SEDDS and Evaluation of the Formulations

Both nonloaded and complex-loaded SEDDS were characterized in terms of size and zeta potential in DI water and MES-HBSS buffer (pH 6.5). The Citrem SEDDS created more uniform nanostructures in terms of size, irrespective of the investigated loading and media. Dispersion of the Standard SEDDS seemed to be more influenced by a loaded substance, as a significant difference in size among the Standard SEDDS formulations was observed (Table 6). However, all SEDDS remained in the range suitable to trigger phagocytosis by macrophages, which are highly active in inflamed intestinal tissue [11,58]. Moreover, in the affected area, positively charged proteins are overexpressed [9]. Thus, negatively charged formulations such as Citrem SEDDS present an attractive approach for local targeting to the diseased tissue.

In DI water, a significant difference in zeta potential between nonloaded and complex-loaded SEDDS was observed for both the Citrem and Standard SEDDS. After complex loading, the zeta potential becomes more positive, suggesting the interference of both cationic lipids from the complexes with the surface charge. Given that OND was complexed with cationic lipids in excess, the shift of zeta potential towards more positive values can be expected. Nevertheless, these differences were not observed in the MES-HBSS buffer (pH 6.5). An interplay between ions present in the buffer and decreased pH seemed to mitigate the differences in zeta potential for individual SEDDS.

The Citrem and Standard SEDDS dispersed in FaSSIF were imaged by cryo-TEM (Figure 7). Previous studies imaging dispersed SEDDS by Tran et al. [59] and Fatouros et al. [60] showed oil droplets after SEDDS dispersion, unlike the Citrem and Standard SEDDS, for which vesicular nanostructures prevailed over oil droplets. In comparison to SEDDS characterized in the previous cryo-TEM studies, Citrem and Standard SEDDS contain a higher percentage of lipophilic surfactants. High concentration of lipophilic surface-active molecules can lead to the formation of vesicles, as well as higher complexity and diversity of the vesicular structures.

The majority of the observed nanostructures were between 100 and 300 nm. This range correlates with the size measured by dynamic light scattering (Table 6).

As the impact of the SEDDS excipients on the complex stability was evaluated, a slight but significant decrease in complex stability induced by anionic Citrem was observed (Table 4). However, this slight decrease seems to have translated into a low ability of the Citrem SEDDS to protect OND. In the presence of the S1 nuclease, only 16% of the loaded OND amount was protected against hydrolysis (Table 7). The Standard SEDDS excipients showed no interference with the complex stability, which improved the protection of the loaded OND 3.5-fold (up to 58% of the loaded OND amount). This suggests that there are also other factors in addition to the negatively charged Citrem that lead to complex destabilization.

Both the Citrem and Standard SEDDS disperse into nanodroplets and vesicular structures (Figure 7). Complexed OND is likely to remain inside the nanodroplets; the vesicle formation, however, could impair the complex stability. It would require further examination to track the complex inside the vesicles and to evaluate the impact of vesicular structures on the stability of the complexes. Thus, OND protection could be further improved by optimization of the SEDDS composition.

Upon oral administration, SEDDS excipients undergo digestion by lipases present in the gastrointestinal tract. The impact of gastric lipases, together with low gastric pH, can be circumvented by administering SEDDS preconcentrates in enteric-coated capsules. In the small intestine, the majority of triglycerides are digested by lipases [61]. In order to maintain the dispersed colloidal structures intact and, thus, to prolong the protection of OND, a lipase inhibitor, orlistat, was added to the SEDDS. Only 0.25% (*w*/*w*) of orlistat sufficed to decrease the level of lipolysis in the tested SEDDS to about 10% (Figure 6). This indicates that orlistat incorporation is a potentially useful approach to inhibit SEDDS digestion and to enable the delivery of well-protected OND. Previous studies have shown that the inhibition of lipolysis of a MCFA-based delivery system did not interfere with its permeation-enhancing effect [62,63].

### 4.3. In Vitro Performance of the Formulations

The ability of the Citrem and Standard SEDDS to enhance the permeation of OND was investigated in a transport study across the Caco-2 cell monolayer. A significant increase in OND in the basolateral compartment of the SEDDS-treated cells correlated with a decrease in TEER. No changes in TEER after 30 min in the Citrem SEDDS treated samples led to a stable P_app_. The TJs opening by the Standard SEDDS plateaued after 60 min, and the final TEER was reached (Figure 8C,D). This resulted in a significantly enhanced permeation for the Standard SEDDS formulations compared to the respective Citrem SEDDS formulations. Finally, all SEDDS formulations enabled a significantly higher amount of transported OND to the basolateral compartment relative to the OND solution (Figure 9). For the Citrem SEDDS, there was no significant difference among the formulations, while the DDAB, DOTAP-loaded and nonloaded Standard SEDDS dispersed in OND solution showed a superior permeation of OND compared to the complex-loaded Standard SEDDS. Due to the complexation, OND is associated within the dispersed SEDDS and, thus, is not readily available for permeation, unlike the case of the noncomplexed OND dissolved in the dispersion medium. Nevertheless, loading OND complexed with a cationic lipid into SEDDS enables the crucial codelivery with PEs into the site of the action [64]. In addition, specific delivery through the intestinal monolayer provided by the Citrem and Standard SEDDS was confirmed, as no fluorescently labeled OND was localized inside the Caco-2 cells after the two-hour-long uptake study (Figure 10). This observation is in accordance with Li et al., who reported only intercellular localization of a macromolecular loading when it was delivered in MCFA-based formulations. In contrast, the loading was observed perinuclearly when employing a formulation based on long-chain fatty acids [65].

The performed experiments did not offer a conclusive answer regarding the choice of a cationic lipid for OND complexation. Although differing in the minimal amount of lipid needed for the complexation (Table 3), as well as the size of the complexes (Figure 3), no differences in the ability to protect OND (Table 7) and in the in vitro performance were observed (Figure 9).

Utilized SEDDS excipients are generally recognized as safe (GRAS) and/or approved by the European and US Pharmacopeia [37,66,67]. All SEDDS disperse into nanoparticles of a comparable size of about 200 nm, but they differ in zeta potential. The addition of Citrem into SEDDS decreases the surface charge of the formed colloidal nanostructures. The negative surface charge diminishes the interactions between SEDDS droplets and the negatively charged cell membranes of the Caco-2 cell monolayer. Due to the lack of this kind of repulsion between the cell surface and Standard SEDDS, there will be more frequent interactions between the Standard SEDDS and the cell membrane, resulting in membrane disturbance and LDH release (Figure 8B). Simultaneously, the opening of TJs by Citrem SEDDS occurs in a less intense manner. Nevertheless, it is noteworthy that McCartney et al. showed the reversible nature of Labrasol-induced reduction in TEER. The same study also reported that a static system such as the Caco-2 monolayer has a limited repair capacity compared to rat colonic mucosae [62].

Medium-chain triglyceride-based SEDDS confirmed their potential in vivo as they outperformed SEDDS based on long-chain triglycerides for the delivery of insulin [63]. However, generally, the SEDDS in vitro–in vivo correlation is known to not always be satisfactory [19], mainly due to the dynamic in vivo environment of complex intestinal fluids of both endogenous and dietary origin and the challenging crossing of the mucus layer [28]. Pathophysiological inflammatory changes, such as the overexpression of positively charged proteins and highly active phagocytes, create conditions for passive targeting. These changes specific to the inflammatory site might favor the negatively charged SEDDS in terms of enhanced effectivity at the target tissue. On the other hand, the permeation enhancement effect of the neutral Standard SEDDS was shown to be more pronounced in the Caco-2 monolayer. The preferential uptake of negatively charged particles by phagocytic cells [68] might limit the ability of the neutral SEDDS to deliver OND into targeted phagocytes.

## 5. Conclusions

The SEDDS based on MCFA was proven to be a viable delivery system across the Caco-2 monolayer for OND drugs. Hydrophobic ion pairing of OND with a cationic lipid, either DDAB or DOTAP, led to the formation of specific complexes, as confirmed by AFM and FTIR. Complexation reduced the hydrophilicity of OND and increased its solubility in lipids. The resulting ion-paired complexes subsequently allowed for their formulation into SEDDS: negatively charged or neutral SEDDS. A morphological evaluation of these SEDDS using cryo-TEM showed the presence of both nanosized oil droplets and vesicular structures. Negatively charged SEDDS did not significantly affect the Caco-2 viability. Additionally, neutral SEDDS enabled a higher OND permeability across the Caco-2 monolayer into the lamina propria, as well as better OND protection against hydrolytic decomposition in comparison to its charged counterpart. The lipolysis of both SEDDS was inhibited with the addition of orlistat, enabling the prolonged protection of OND in the gastrointestinal tract. The evaluation of further properties of the tested formulations, e.g., passive targeting and their effect on intestinal histology, in an appropriate in vivo model could reveal more details of this potent delivery system for local OND delivery. 

## Figures and Tables

**Figure 1 pharmaceutics-13-00459-f001:**
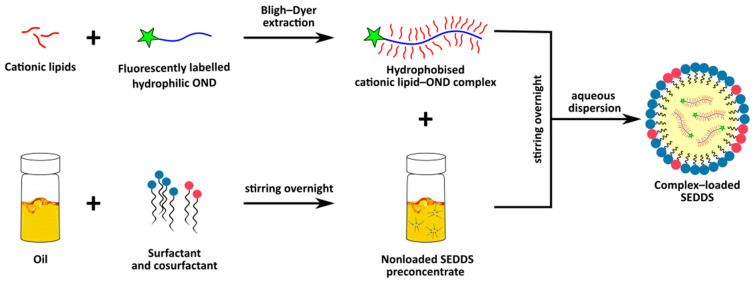
A general scheme showing the formulation of an oligonucleotide (OND) into a self-emulsifying drug delivery system (SEDDS).

**Figure 2 pharmaceutics-13-00459-f002:**
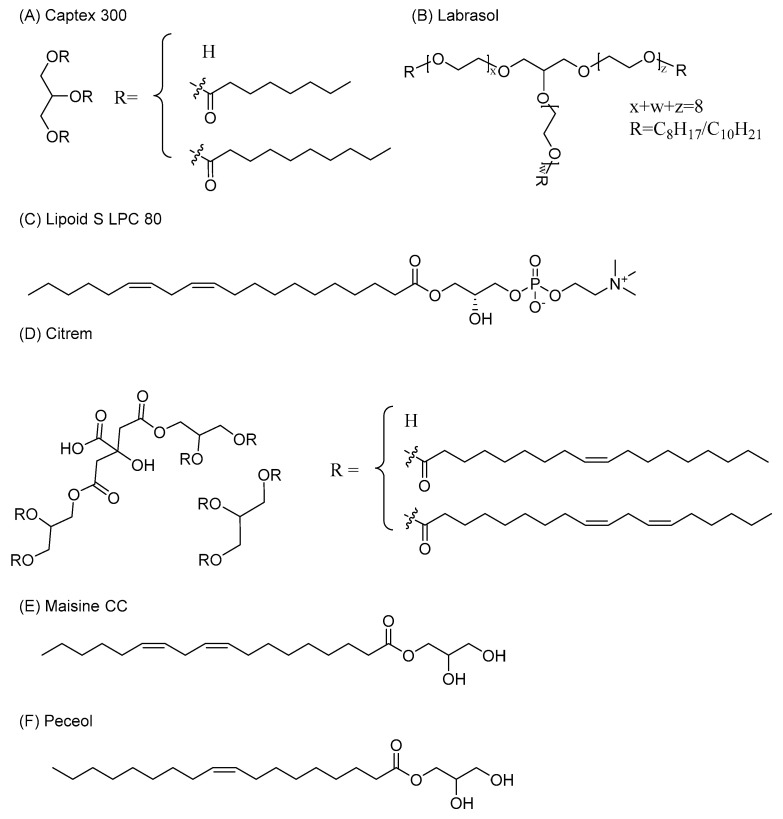
Chemical structures of SEDDS excipients (the structure of Labrasol was adapted from [36], and the structure of Citrem was adapted from [37]).

**Figure 3 pharmaceutics-13-00459-f003:**
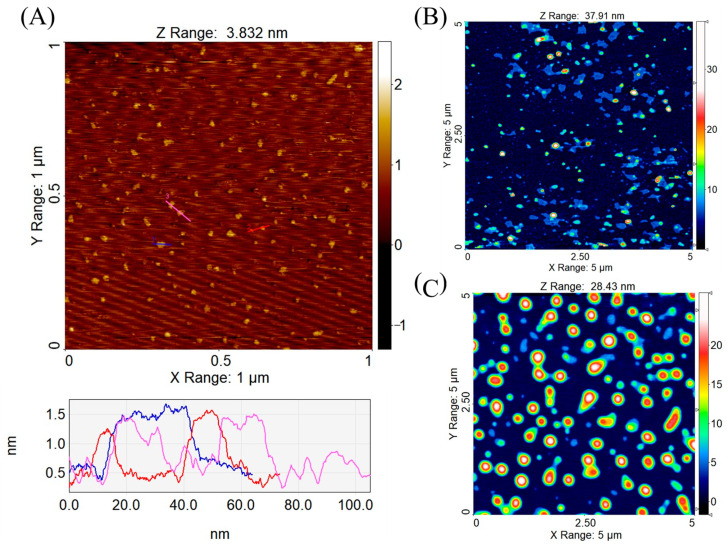
Atomic force microscopy (AFM) topography images of structures absorbed in air on mica: (**A**) OND—the height profile corresponds to three colored sections in the graph below the image, (**B**) dimethyldioctadecylammonium bromide (DDAB)-OND complex and (**C**) 1,2-dioleoyl-3-trimethylammonium propane (DOTAP)-OND complex, with both complexes prepared at a charge ratio 3:1 (cationic lipid:OND).

**Figure 4 pharmaceutics-13-00459-f004:**
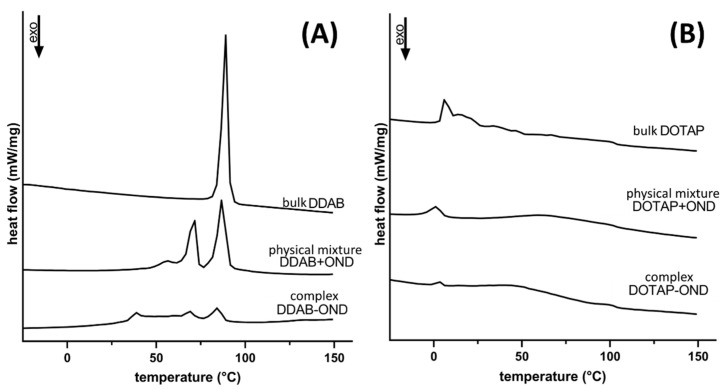
Differential scanning calorimetry of the bulk lipids, physical mixtures and complexes of (**A**) DDAB and (**B**) DOTAP. Thermograms were recorded from −50 up to 150 °C at the heating rate 10 °C/min.

**Figure 5 pharmaceutics-13-00459-f005:**
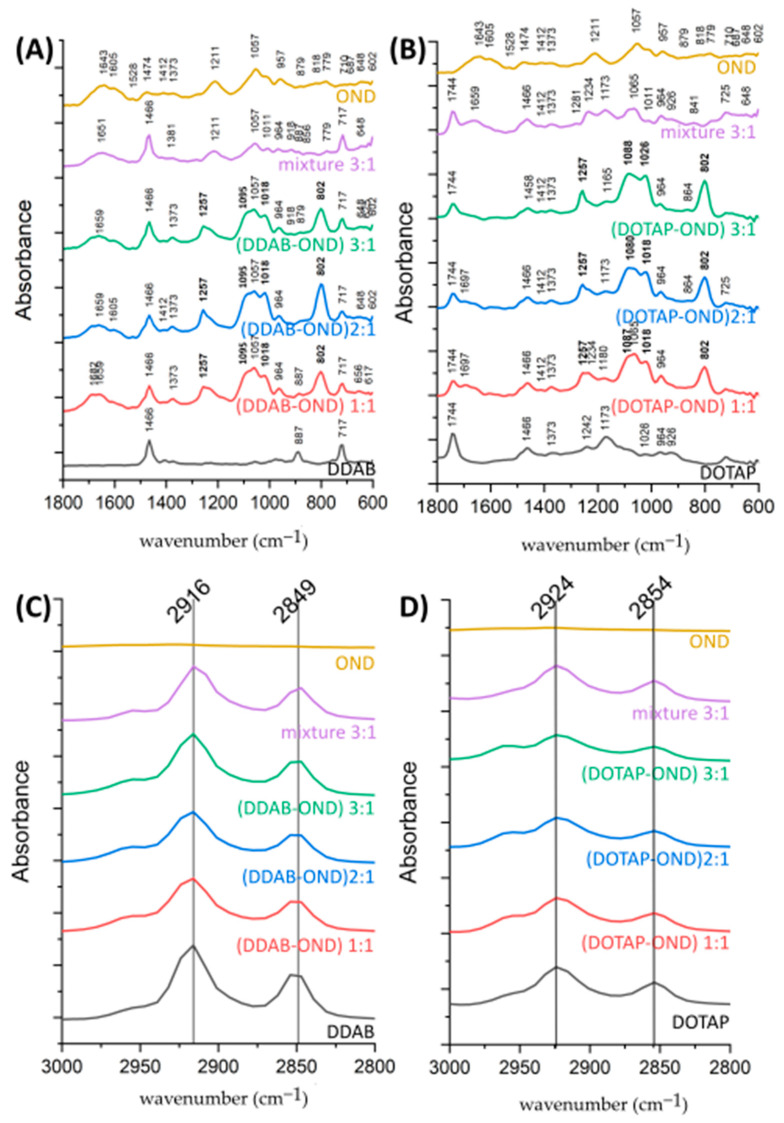
Attenuated total reflectance-Fourier-transform infrared (ATR-FTIR) absorbance spectra of OND, cationic lipid-OND complexes at various ratios, a cationic lipid-OND physical mixture at the ratio 3:1 and a bulk cationic lipid. Spectra of the OND-specific region (1800–600 cm^−1^) of DDAB-OND (**A**) and DOTAP-OND (**B**), and the spectra of the lipid CH2 stretching vibrations (3000–2800 cm^−1^) of DDAB-OND (**C**) and DOTAP-OND (**D**).

**Figure 6 pharmaceutics-13-00459-f006:**
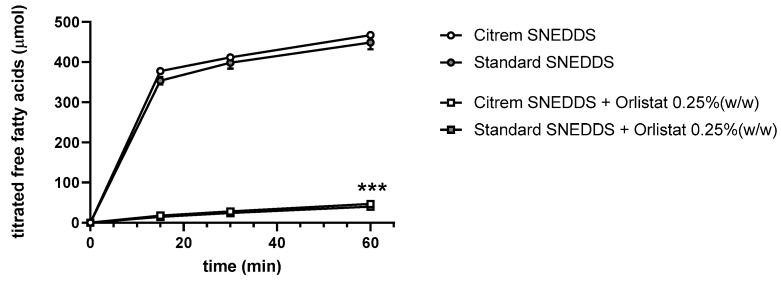
In vitro profile of digestion of SEDDS in fasted-state simulated intestinal fluid presented as the amount of free fatty acids titrated by sodium hydroxide over 60 min. Data are presented as mean ± SD (*n* = 3). *** *p* < 0.001—significant effect of orlistat on SEDDS lipolysis.

**Figure 7 pharmaceutics-13-00459-f007:**
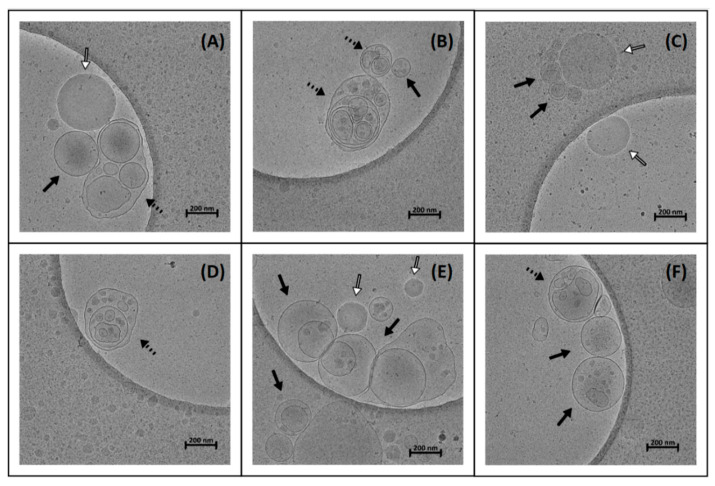
Cryo-TEM images of SEDDS, dilution 1:100 (*v*/*v*) in fasted-state simulated intestinal fluid (FaSSIF). (**A**) Nonloaded Citrem SEDDS, (**B**) DDAB-OND-loaded Citrem SEDDS (100 nmol/g), (**C**) DOTAP-OND-loaded Citrem SEDDS (100 nmol/g), (**D**) nonloaded Standard SEDDS, (**E**) DDAB-OND-loaded Standard SEDDS (100 nmol/g) and (**F**) DOTAP-OND-loaded Citrem SEDDS (100 nmol/g). Full arrows indicate uni- and multilamellar vesicles, dashed arrows indicate more complex vesicular structures and open arrows indicate oil droplets.

**Figure 8 pharmaceutics-13-00459-f008:**
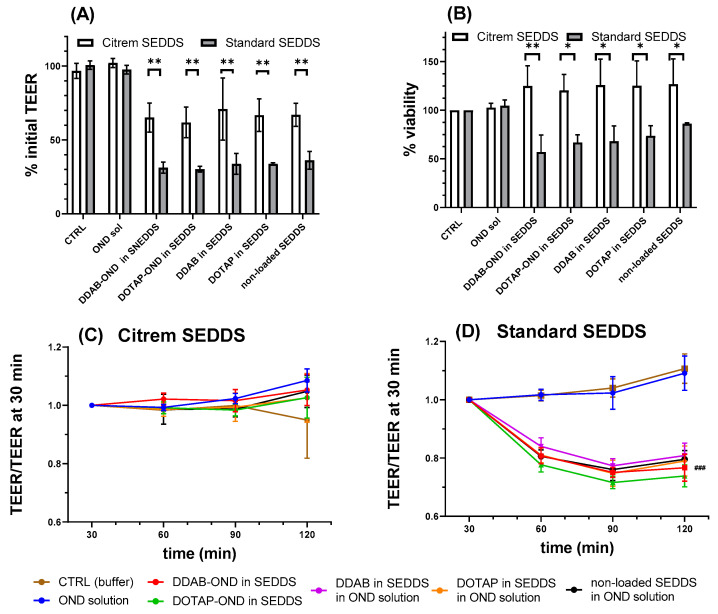
(**A**) Percent of initial transepithelial electrical resistance (TEER) measured by Endohm chamber after 120 min of incubation in both Citrem and Standard SEDDS. ** denotes a significant difference of TEER decline (*p* < 0.01) between the Citrem and Standard SEDDS. All SEDDS-incubated cells showed significantly lower TEER relative to the respective CTRL (buffer) and OND solution. (**B**) Percent of viability by lactate dehydrogenase (LDH) assay after 120 min of incubation with Citrem and Standard SEDDS. * and ** denote significant differences in the viability between different SEDDS (* *p* <0.05 and ** *p* < 0.01). (**C**,**D**) Relative TEER values measured by chopstick electrodes at 37 °C for Citrem SEDDS (**C**) and Standard SEDDS (**D**). (**D**) ^###^ represents a significant difference of the relative TEER values of SEDDS-treated cells relative to the CTRL and OND solution (*p* < 0.001). Results are presented as mean ± SD (*n* = 5 to 6).

**Figure 9 pharmaceutics-13-00459-f009:**
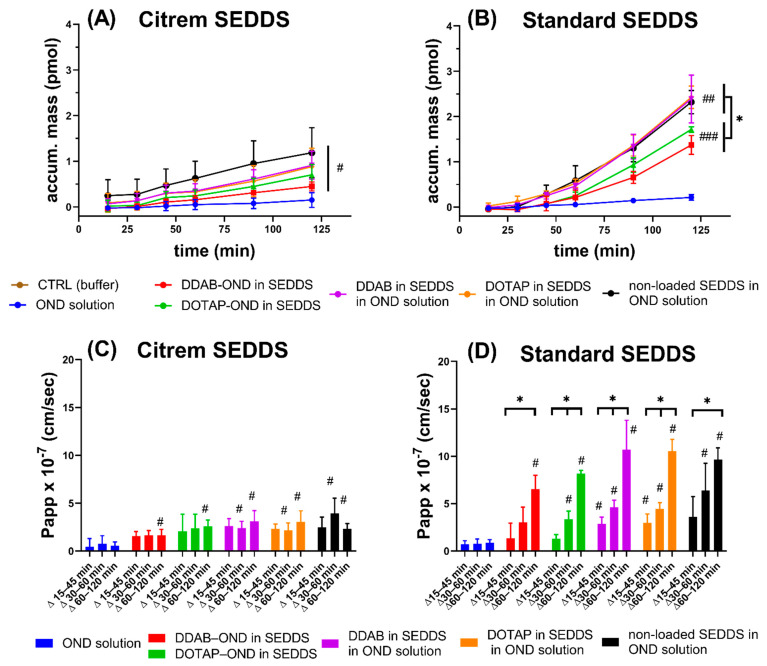
The transported OND accumulated in the basolateral compartment for Citrem SEDDS (**A**) and Standard SEDDS (**B**). The apparent permeability coefficient (P_app_) values for specified time intervals in the Citrem SEDDS (**C**) and Standard SEDDS (**D**). (**A**,**B**) ^#^, ^##^ and ^###^ represent significant differences of the flux curves of SEDDS-treated cells relative to the OND solution (^#^
*p* <0.05, ^##^
*p* < 0.01 and ^###^
*p* < 0.001). (**B**) * represents a significant difference between complex-loaded Standard SEDDS and a group of other Standard SEDDS formulations (* *p* < 0.05). (**C**,**D**) Bars marked by ^#^ showed a significant difference in the P_app_ relative to the respective bars of the OND solution (*p* < 0.05). (**D**) The significant differences between the time intervals in the Standard SEDDS is denoted as * *p* < 0.05. The results are presented as mean ± SD (*n* = 5 to 6).

**Figure 10 pharmaceutics-13-00459-f010:**
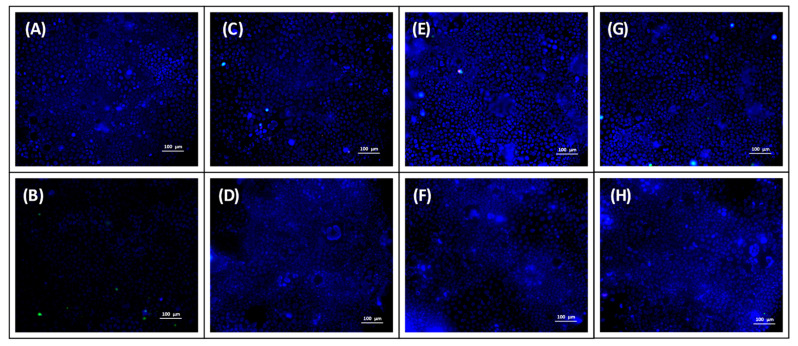
Fluorescence microscope images of fluorescently labeled oligonucleotide (OND) (**A**) MES-HBSS buffer, (**B**) OND solution in MES-HBSS buffer, (**C**,**D**) nonloaded Citrem and Standard SEDDS dispersed in OND solution, respectively, (**E**,**F**) DDAB-OND loaded in Citrem and Standard SEDDS, respectively, and (**G**,**H**) DOTAP-OND loaded in Citrem and Standard SEDDS, respectively. The size bar represents 100 µm. Blue represents the cell nuclei of living cells stained by Hoechst dye, red represents dead cells stained by propidium iodide and green represents fluorescently labeled OND.

**Table 1 pharmaceutics-13-00459-t001:** Composition of the self-emulsifying drug delivery systems (SEDDS).

SEDDS	Captex 300	Labrasol	Lipoid S LPC 80	Citrem	Maisine CC	Peceol
	(%*w*/*w*)	(%*w*/*w*)	(%*w*/*w*)	(%*w*/*w*)	(%*w*/*w*)	(%*w*/*w*)
Citrem (negatively charged)	20	40	20	20	-	-
Standard (neutral)	20	40	20	-	10	10

**Table 2 pharmaceutics-13-00459-t002:** The loaded self-emulsifying drug delivery systems (SEDDS).

Name of Formulation	Loaded Substance	Concentration of Loaded Substance in SEDDS
DDAB-OND in SEDDS	Complex of DDAB and OND at the charge ratio 3:1	100 nmol of OND/g SEDDS
DOTAP-OND in SEDDS	Complex of DOTAP and OND at the charge ratio 3:1	100 nmol of OND/g SEDDS
DDAB in SEDDS	DDAB	3.8 mg/g SEDDS
DOTAP in SEDDS	DOTAP	4.2 mg/g SEDDS
SEDDS + Orlistat	Orlistat	0.25%(*w*/*w*)

DDAB: dimethyldioctadecylammonium bromide, DOTAP: 1,2-dioleoyl-3-trimethylammonium-propane (chloride salt) and OND: oligonucleotide.

**Table 3 pharmaceutics-13-00459-t003:** Complexation efficiency (CE, %) of OND with a cationic lipid (DDAB or DOTAP) using the Bligh–Dyer method. Various charge ratios between a quaternary ammonium head group of a cationic lipid (N+) and a nucleotide phosphate group in the backbone of 20-mer OND (PO_2_^−^) were tested.

Charge Ratio N^+^: PO_2_^−^	Molar Ratio Cationic Lipid: OND	CE (%)	
		DDAB	DOTAP
1:1	20:1	58.3 ± 0.7	84.6 ± 1.4
2:1	40:1	84.0 ± 2.8	100.1 ± 0.1
3:1	60:1	96.5 ± 2.0	100.4 ± 0.1

Results are presented as mean ± SD (*n* = 3).

**Table 4 pharmaceutics-13-00459-t004:** Effect of SEDDS excipients on the complex stability. The complexation efficiency (CE %) after interactions with a SEDDS excipient is reported relative to the control where no SEDDS excipient was added (CE = 100%).

SEDDS Excipient	CE (%)	
	DDAB-OND	DOTAP-OND
Citrem	97.3 ± 0.2 ***	97.7 ± 0.2 ***
LPC	100.0 ± 0.2	n.d.
Captex 300	100.3 ± 0.1	100.1 ± 0.1
Labrasol	100.5 ± 0.2	100.2 ± 0.1
Maisine CC	100.8 ± 0.1	100.4 ± 0.1
Peceol	100.2 ± 0.4	100.3 ± 0.1

*** statistically significant difference relative to a control with added lipid. Results are presented as mean ± SD (*n* = 3). n.d. = not determined.

**Table 5 pharmaceutics-13-00459-t005:** Bands specific to the complexes of both cationic lipids and OND. Fourier-transform infrared (FTIR) spectra of the respective cationic lipid and OND were subtracted from the FTIR spectra of the complexes to obtain complex-specific bands. Bands were observed in the complexes at all ratios if not otherwise specified.

Complex-Specific Bands (cm^−1^)	Marker Bands Characteristic of	Assignment	Reference
1257	Phosphate-deoxyribose backbone	Organic phosphate P=O, vibrational asymmetric band	[24,46,48,49]
1095 DDAB-OND 1088-1080 DOTAP-OND	Phosphate-deoxyribose backbone	Organic phosphate P=O, vibrational symmetric band	[24,46,49]
1018	Phosphate-deoxyribose backbone	P-O-C aliphatic phosphate	[48]
802	Deoxyribose conformation	N-type (C3′-endo) puckering mode	[46]

**Table 6 pharmaceutics-13-00459-t006:** Size and zeta potential of Citrem and Standard SEDDS. The parameters were tested in deionized (DI) water and in 2-(N-morpholino)ethansulfonic acid-Hanks’ Balanced Salt Solution (MES-HBSS) (pH 6.5). PdI: polydispersity index.

		Size (nm)		PdI		Zeta Potential (mV)	
SEDDS	Loaded Substance	DI Water	MES-HBSS	DI Water	MES-HBSS	DI Water	MES-HBSS
Citrem	nonloaded	201 ± 11	207 ± 16	0.36	0.27	−35.5 ± 0.6	−10.9 ± 0.7
	DDAB-OND	209 ± 14	237 ± 11	0.30	0.23	−24.1 ± 0.8 ^###^	−9.9 ± 0.7
	DOTAP-OND	195 ± 19	213 ± 5	0.30	0.22	−26.5 ± 1.2 ^###^	−9.4 ± 0.5
Standard	nonloaded	223 ± 10	213 ± 44	0.17	0.22	−5.2 ± 2.1	−1.2 ± 0.5
	DDAB-OND	256 ± 22	267 ± 10 ***	0.28	0.12	13.0 ± 1.3 ^###^	0.2 ± 0.5
	DOTAP-OND	240 ± 9 *	183 ± 27	0.23	0.25	14.0 ± 0.9 ^###^	0.3 ± 0.3

* *p* < 0.05 statistical difference relative to nonloaded Standard SEDDS in DI water, *** *p* < 0.001 statistical difference relative to nonloaded Standard SEDDS in MES-HBSS buffer and ^###^
*p* < 0.001 statistical difference relative to the respective nonloaded SEDDS. Results are presented as mean ± SD (*n* = 3).

**Table 7 pharmaceutics-13-00459-t007:** Percentage of OND protected from degradation after 30 min of incubation in the presence of the S1 nuclease.

SEDDS	Intact OND (%)	
	DDAB-OND in SEDDS	DOTAP-OND in SEDDS
Citrem	16.0 ± 1.5	15.7 ± 1.0
Standard	59.9 ± 3.4 ***	57.1 ± 3.0 ***

*** *p* < 0.001 is the statistically significant difference between Citrem and Standard SEDDS. Data are presented as mean ± SD (*n* = 3).

**Table 8 pharmaceutics-13-00459-t008:** Transported OND accumulated in the basolateral compartment after 120 min presented as the total mass (pmol) and % of the initial apical mass.

Formulation	Citrem SEDDS		Standard SEDDS	
	Transported OND Accumulated Basolaterally at 120 min		Transported OND Accumulated Basolaterally at 120 min	
	pmol	%	pmol	%
OND solution	0.15 ± 0.16	0.03	0.21 ± 0.06	0.04
DDAB-OND in SEDDS	0.45 ± 0.10 *	0.09	1.37 ± 0.21 **	0.28
DOTAP-OND in SEDDS	0.70 ± 0.23 ***	0.14	1.72 ± 0.06 ***	0.34
DDAB in SEDDS	0.91 ± 0.31 **	0.18	2.40 ± 0.53 **	0.48
DOTAP in SEDDS	0.88 ± 0.41 **	0.17	2.43 ± 0.25 ***	0.49
Nonloaded SEDDS	1.19 ± 0.55 **	0.24	2.32 ± 0.26 ***	0.46

Results are presented as mean ± SD (*n* = 5 to 6). * *p* < 0.05, ** *p* < 0.01 and *** *p* < 0.001 = statistically significant differences in transported OND formulated into SEDDS in comparison to the OND solution.

## Data Availability

The data is available on reasonable request from the corresponding author.

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
