# Peer review of "Oligonucleotide Delivery across the Caco-2 Monolayer: The Design and Evaluation of Self-Emulsifying Drug Delivery Systems (SEDDS)"

_pharmaceutics, 2021, doi:10.3390/pharmaceutics13040459_

Round 1
Reviewer 1 Report
- The authors are suggested to thoroughly go through the manuscript and fix all the format errors to assist the Reviewers to evaluate. The current manuscript has "Error! Reference source not found" shown at least 9 times. Figure 2 showed up 5 times in the manuscript. All these errors made the work hard to assess.
- Please add a general scheme to describe how the delivery system is formulated.
- Experimental methods are not described in detail. For example, Line 208 says "The SEDDS were loaded by dissolving the payload under stirring at 37 °C overnight..." It might be due to the language issue of the manuscript too, but how was the dissolvation conducted (volume, solvent, concentrations, etc. )? What exactly is the payload? Likewise, please revise the rest of the "Materials and Methods" section.
- What is the difference between the "physical mixture" and "complexation"? It should be clearly described. Do they have difference in the lipid/OND ratio?
- Please provide the structure of Citrem, LPC, Captex 300, Labrasol, Maisine CC and Peceol.
- Where is the toxicity data described in 3.11?
- The authors stated that "random DNA nucleotides" was used for the study. If the formulation does not result in any phenotype from the OND, what is the importance of doing OND delivery?
Author Response
Dear Reviewer,
On behalf of my co-authors, I would like to thank you for your constructive and insightful feedback on our manuscript. We found the feedback valuable and helpful in revising our manuscript and we carefully considered and addressed the provided comments.
First of all, we would like to sincerely apologize for the format problem that unexpectedly arose (likely in the process of conversion from docx into the pdf format). It has been resolved.
Replies to reviewer's comments:
“Please add a general scheme to describe how the delivery system is formulated.”
Thank you for the valuable suggestion. We added the scheme into revised manuscript. Please see Figure 1
“Experimental methods are not described in detail. For example, Line 208 says "The SEDDS were loaded by dissolving the payload under stirring at 37 °C overnight..." It might be due to the language issue of the manuscript too, but how was the dissolvation conducted (volume, solvent, concentrations, etc. )? What exactly is the payload? Likewise, please revise the rest of the "Materials and Methods" section.”
The entire section 2.7 (Line 215-241) has been revised, and the preparation of complexes was further specified. Terms used in Table 2 were adjusted to clarify the preparation. Material and Method section was thoroughly checked, and details were provided.
“What is the difference between the "physical mixture" and "complexation"? It should be clearly described. Do they have difference in the lipid/OND ratio?”
Section 2.2 has been adopted and specification of utilized ratio was added throughout the manuscript. Please see Line 159-165 and 689-690
“Please provide the structure of Citrem, LPC, Captex 300, Labrasol, Maisine CC and Peceol.”
The figure of structures has been added. Please see Figure 2.
“Where is the toxicity data described in 3.11?”
The data are shown in Figure 8B; line 644
“The authors stated that "random DNA nucleotides" was used for the study. If the formulation does not result in any phenotype from the OND, what is the importance of doing OND delivery?”
This issue is discussed and explained in Lines 102-104 and in Lines 659-664 of our revised manuscript.
Yours sincerely,
Ondrej Holas
Reviewer 2 Report
In this manuscript, authors prepared and characterized SEDDS for delivery of oligonucleotides (OND) and compared negatively charged SEDDS and neutral SEDDS on OND protection and Caco-2 permeability, and damage of Caco-2 cells. The experiments were done properly and the manuscript is basically well written. But the revision is needed for the manuscript. I have several comments.
The structure of the manuscript seems broken. For example, many same figures (Figure 2) are included (page 11, 12, 20). Where is Figure 3? Correct them.
The composition of SEDDS used in this study is shown in Table 1. Authors should explain why the composition was used in manuscript as data of Ramakrishnan et al. were not published (line 204).
Where is the data of in vitro toxicity study (lines 621-629)? If it is not shown in manuscript, “data not shown” should be written.
Authors should discuss the pros and cons of negatively charged SEDDS and neutral SEDDS for in vivo application.
Minor comments.
Lines 91, 92. The sentence seems grammatically incorrect.
Author Response
Dear Reviewer,
On behalf of my co-authors, I would like to thank you for your constructive and insightful feedback on our manuscript. We found the feedback valuable and helpful in revising our manuscript and we carefully considered and addressed the provided comments.
First of all, we would like to sincerely apologize for the format problem that unexpectedly arose (likely in the process of conversion from docx into the pdf format).
Replies to reviewer's comments:
“The structure of the manuscript seems broken. For example, many same figures (Figure 2) are included (page 11, 12, 20). Where is Figure 3? Correct them.”
We apologize for the formatting issue; it has now been fixed.
“The composition of SEDDS used in this study is shown in Table 1. Authors should explain why the composition was used in manuscript as data of Ramakrishnan et al. were not published (Line 204).”
The issue is addressed on Lines 221-223
“Where is the data of in vitro toxicity study (Lines 621-629)? If it is not shown in manuscript, “data not shown” should be written.”
The data are shown in Figure 8B; Line 644
“Authors should discuss the pros and cons of negatively charged SEDDS and neutral SEDDS for in vivo application.”
Please see the last two paragraphs of the discussion: Lines 778-799
“Lines 91, 92. The sentence seems grammatically incorrect.”
The sentence was rewritten, please see Lines 102-104
Yours sincerely,
Ondrej Holas
Round 2
Reviewer 1 Report
The authors have addressed most of the comments from previous reviewers. One missing experimental evidence for the study is the cellular uptake and transportation of these formulations in Caco-2 cells. As the authors mentioned in Line 654 of the revised manuscript that non-specific OND is easily detectable, it should be proved with fluorescence microscopy results.
Author Response
Dear reviewer,
thank you for your valuable comment on the manuscript. We completely agree that fluorescent microscopy provide a valuable evidence regarding a permeation enhancement mechanism. The cell uptake study was conducted, and fluorescent microscopy images have been added. Please see lines 362-383 for materials and methods of uptake study; lines 674-691 and Figure 10 for cell uptake study results; lines 812-818 for the results discussion. Supplementary data showing individual fluorescent microscopy images is added to the submission.
We have also had the manuscript checked by native speaker in order to make the manuscript as understandable as possible.
Sincerely yours,
Ondrej Holas
Reviewer 2 Report
The manuscript is considerably improved after revision.
Functional data of introduced OND are interesting and should be added in the next study. I also think that it is better to add fluorescence microscopy results in this study as the experiments were conducted with a model fluorescently labelled OND.
English should be further improved through manuscript for general readers to be able to read more easily.
Line 769. GRAS stands for “generally recognized as safe”, not “generally considered as safe”.
The structure of sentence of Lines 790-793 seems incorrect. It should be corrected.
Author Response
Dear reviewer,
thank you for your insightful comment on the manuscript. We completely agree that fluorescent microscopy provide a valuable evidence regarding a permeation enhancement mechanism. The cell uptake study was conducted, and fluorescent microscopy images have been added. Please see lines 362-383 for materials and methods of uptake study; lines 674-691 and Figure 10 for cell uptake study results; lines 812-818 for the results discussion. Supplementary data showing individual fluorescent microscopy images is also available.
The sentence previously on the lines 790-793 was adopted. Please see lines 845-849.
Thank you for the remark regarding GRAS abbreviation, it has been corrected.
We have also had the manuscript checked and re-checked by native speaker to make this manuscript as understandable as possible.
Sincerely yours,
Ondrej Holas